# Clinical predictors of radiological pneumonia: A cross-sectional study from a tertiary hospital in Nepal

**Sandeep Shrestha**[1], **Nagendra Chaudhary**[1]*, **Saneep Shrestha**[2], **Santosh Pathak**[3], **Arun Sharma**[4], **Laxman Shrestha**[4], **Om P. Kurmi**[5]

1 Department of Pediatrics, Universal College of Medical Sciences, Bhairahawa, Nepal, 2 Department of Community Medicine, Universal College of Medical Sciences, Bhairahawa, Nepal, 3 Department of Pediatrics, Chitwan Medical College, Bharatpur, Nepal, 4 Department of Pediatrics, Tribhuvan University Teaching Hospital, Institute of Medicine, Kathmandu, Nepal, 5 Division of Respirology, Department of Medicine, McMaster University, Hamilton, Canada

* enagendra@hotmail.com

## Abstract

### Background

Despite readily availability of vaccines against both *Hemophilus influenzae* and *Pneumococcus*, pneumonia remains the most common cause of morbidity and mortality in children under the age of five years in Nepal. With growing antibiotic resistance and a general move towards more rational antibiotic use, early identification of clinical signs for the prediction of radiological pneumonia would help practitioners to start the treatment of patients. The main aim of this study was to reassess the clinical predictors of pneumonia in Nepal.

### Methods

This cross-sectional study was conducted between June 2015 and November 2015 at Tribhuvan University Teaching Hospital, a tertiary hospital in Kathmandu, Nepal. Children aged 3–60 months with a clinical diagnosis of pneumonia by a physician were enrolled in the study. Radiological pneumonia was identified and categorized as per World Health Organization guidelines by an experienced radiologist blinded to patient characteristics. We calculated sensitivity and specificity of clinical signs and symptoms for radiological pneumonia.

### Results

Out of 1021 children with fever, 160 cases were clinically diagnosed as pneumonia and were enrolled for this study. Among the enrolled patients, 61% had radiological pneumonia. Tachypnea had the highest sensitivity of 99%, while bronchial breathing had the highest specificity of 100%. During univariate analysis, grunting, wheezing, nasal discharge, decreased breath sounds, noisy breathing and hypoxemia were associated with radiological pneumonia. Only hypoxemia remained an independent predictor when adjusted for all the factors.

**Data Availability Statement:** All relevant data are within the manuscript and its Supporting Information files.

**Funding:** The authors received no specific funding for this work.

**Competing interests:** The authors have declared that no competing interests exist.

## Conclusion

Tachypnea was the most sensitive sign, whereas bronchial breathing was most specific sign for radiological pneumonia.

## Introduction

Pneumonia is one of the most common causes (followed by prematurity related deaths) of childhood infections attributed to about 2 million children deaths worldwide [1]. The diagnosis of pneumonia in children remains an important yet difficult clinical problem, particularly in resource poor setting. Although fast breathing has been recommended as a predictor of childhood pneumonia, no clinical sign can solely predict pneumonia [2].

The World Health Organization (WHO) uses tachypnea (age 2–11 months, ≥50/min; age 1–5 years, ≥40/min) and/or lower chest indrawing as a sole criterion to diagnose pneumonia in children with a cough or breathing difficulty [3]. In low- and middle-income countries (LMICs), including Nepal, chest x-ray usually remains the diagnostic test of choice and often, health workers, including treating physicians, use WHO guidelines to diagnose and treat pneumonia [4]. Due to difficulty in obtaining appropriate specimens from the lower respiratory tract for culture and microbiological evaluation, radiography has been considered as the best method available for diagnosing pneumonia [4–6]. However, there is still a dilemma regarding when to order a chest x-ray in a case of suspected pneumonia. Earlier studies reported the clinical predictors of radiological pneumonia when the cases associated with radiological pneumonia were usually caused by bacterial agents mostly *Streptococcus pneumoniae* and *Hemophilus influenzae* [7–9]. WHO guidelines developed earlier for the detection and management of childhood pneumonia targeted bacterial agents [3, 10].

Currently, with the introduction of *Hemophilus influenzae type b* (Hib) and pneumococcal conjugate vaccines (PCV) and global expansion of their coverage, bacterial agents are on the decline and out-numbered by viral and atypical bacteria [11, 12]. The clinical presentation and the radiographic signs of pneumonia may not be the same as found earlier. Considering the change in the epidemiological pattern, the clinical predictors of pediatric pneumonia need reassessment. Hence, this hospital-based study was conducted to find the predictors of radiological pneumonia.

## Methods

### Study design, hospital setting, participants and diagnosis

This cross-sectional study was conducted from June 2015 to November 2015 in Tribhuvan University Teaching Hospital (TUTH), a tertiary healthcare centre in Kathmandu, Nepal, which lies at an altitude of 1400 meters (4600 ft) from the sea level. Children aged 3–60 months who visited the TUTH at an outdoor patient department or emergency unit and presented with fever, cough, and difficulty or fast breathing were enrolled in this study. All the clinical pneumonia cases included were community-acquired (CAP). No hospital-acquired pneumonia cases were included in the study. Children with pre-existing cardiac disease, chronic respiratory disease (cystic fibrosis/bronchopulmonary dysplasia), known asthma or presenting with asthma (requiring >1 bronchodilator or systemic steroids), history of foreign body aspiration, history of receiving antibiotics >1 week and with chest x-ray outside TUTH (in private clinics)

were excluded from the study. Etiological diagnosis (bacterial versus viral) was not preformed in the present study.

Parents of all children enrolled in this study provided vaccination history details, including *Hib* and *PCV* vaccines. The youngest child included in this study was three months and had received at least one dose of *Hib* and *PCV-10* vaccines. A respiratory physician examined the children for the presence of tachypnea, nasal flaring, grunting, chest indrawing, decreased air entry, bronchial breath sounds and hypoxemia. Experienced medical officers from emergency departments or senior pediatric residents blinded to radiological findings of the children, screened them at the out-patient clinic.

## Study definitions and variables

WHO cut-off points were taken to define age-adjusted tachypnea- children between 2–11 months (50 or more breaths/min) and 12–59 months (40 or more breaths/min) [3]. Clinical pneumonia was defined as a child having a fever, cough, difficult and/or fast breathing. Hypoxemia was defined as oxygen saturation less than 90% in the pulse oximeter (Mini $SPO_2$, Criticare Systems, USA) measured by the pediatric probe.

Fever was defined as an axillary temperature of 100.4 ° F or more. All x-rays were carried out using the same portable digital x-ray machine (SHEMADZU 500mA Shandong, Mainland China). All x-ray films were interpreted by an experienced radiologist, blinded to the clinical features of the child's condition. The presence of consolidation, asymmetrical infiltrates, or air bronchograms was considered as radiological pneumonia. A diagnostic agreement was made between the evaluating pediatrician and radiologist in all cases.

## Study outcomes

Clinical pneumonia was categorised as radiological and non-radiological pneumonia based on the x-ray findings. The sensitivity and specificity of each of the clinical predictors were then calculated.

## Ethical approval

The study was approved by the Institute's Research Committee (IRC) of Tribhuvan University Teaching Hospital (TUTH), Institute of Medicine, Kathmandu, Nepal *[reference number-37 (6-11-0)], dated 26th August 2014]*. Written and oral consent were obtained from the parents.

## Statistical analysis

Descriptive statistics were used to report the characteristics of all children enrolled in this study. The heterogeneity between different baseline characteristics for children with radiological versus non-radiological pneumonia were tested using Chi-square test for categorical variables and t-test for continuous variables. We calculated the crude odds ratio for clinical signs and symptoms of radiological pneumonia using regression analysis. A multivariable regression analysis was carried out after adjusting for all the clinical signs and symptoms. SPSS software (*version 21.0 IBM, Armonk, NY, USA*) was used for data entry and analysis. Sensitivity and specificity of each variable for radiological pneumonia were calculated.

## Results

Out of 4211 children visiting the out-patient and emergency unit of the pediatric department of TUTH, 1021 patients had a fever and were screened for the presence of clinical pneumonia

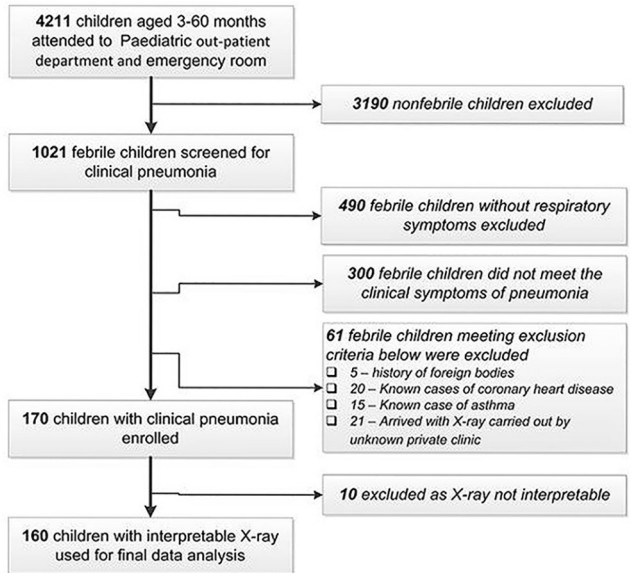

**Fig 1. Flow chart showing the selection of pneumonia participants.**

(fever, cough and fast or difficulty breathing). Fig 1 shows the flow diagram of the enrolled children in this study.

Out of a total of 160 children enrolled, 68% were males, and about 60.6% had radiological pneumonia. Table 1 shows the baseline characteristics of children with radiological pneumonia and non-radiological pneumonia. The proportion of children with radiological pneumonia was greater in both males (59.6% versus 40.4%) and females (62.7% versus 37.3%) children when compared to non-radiological pneumonia (Table 1). Oxygen saturation (<90%) and high total leukocyte counts were found to be significantly associated with radiological pneumonia.

Children with radiological pneumonia had complications such as empyema, myocarditis, parapneumonic effusion, respiratory failure and septic shock. The common signs and symptoms present in the enrolled children are reported in Figs 2 and 3, respectively. Noisy breathing and refusal to feeds were common clinical presentation in children with pneumonia and was predominant in children ≤12 months (Fig 2). On examination, tachypnea (99% in 3–12 months and 96% in 13–60 months), crepitation (75% in 3–12 months and 71% in 13–60 months), retraction (72% in 3–12 months and 45% in 13–60 months) and hypoxemia (68% in 3–12 months and 51% in 13–60 months) were common clinical signs noticed and was predominantly more in children aged 3–12 months (Fig 3).

Table 2 shows the statistical comparison of clinical features between children with and without radiological pneumonia. Noisy breathing (p = 00.2) and nasal discharge (p = 0.02) were the clinical symptoms which were significantly associated with radiological pneumonia. The sensitivity and specificity of noisy breathing were 44.3% and 30.2% respectively whereas for nasal discharge were 15.5% and 69.8% respectively. Among the clinical signs, grunting (p = 0.044), hypoxemia (p = 0.005), wheezing (p<0.001), decreased breath sounds (p<0.001), and bronchial breath sounds were significantly associated with radiological pneumonia in the children. No significant association of radiological pneumonia was observed with tachypnea, nasal flaring, retraction and crepitation (Table 2). Among various clinical variables, age-adjusted tachypnea had the highest sensitivity (99%) with low specificity (6.35%). Grunting

**Table 1. Baseline characteristics of children enrolled with and without radiological pneumonia.**

| Parameters | Radiological pneumonia (Mean ± SE) or [n (%)] | Non-radiological pneumonia (Mean ± SE) or [n (%)] | p-value |
|---|---|---|---|
| N | 97 (60.6) | 63 (39.4) | |
| Age (months) | 22.6 ± 1.8 | 21.4 ± 2.3 | 0.339 |
| Gender | | | |
| Male | 65 (59.6) | 44 (40.4) | 0.707 |
| Female | 32 (62.7) | 19 (37.3) | |
| Birth weight (kg) | 2.8 ± 0.04 | 3.0 ± .0.05 | 0.019 |
| Height for age (z-score) | | | |
| +2 to +3 | 14 (60.9) | 9 (39.1) | 0.927 |
| 0 to +2 | 38 (58.5) | 27 (41.5) | |
| 0 to -2 | 34 (64.2) | 19 (35.8) | |
| -2 to -3 | 11 (57.9) | 8 (42.1) | |
| Weight for height | | | |
| +2 to +3 | 4 (50.5) | 4 (50.0) | 0.346 |
| 0 to +2 | 21 (55.3) | 17 (44.7) | |
| 0 to -2 | 64 (66.0) | 33 (34.0) | |
| -2 to -3 | 8 (47.1) | 9 (52.9) | |
| History of family smoking (Yes) | 7 (63.6) | 4 (36.4) | 0.832 |
| Temperature (deg F) | 100.6 ± 1.1 | 100.7 ± 0.1 | 0.95 |
| Respiratory rate (per min) | 60.1 ± 1.0 | 58.7 ± 1.7 | 0.232 |
| Oxygen saturation (%) | 88.3 ± 0.5 | 90.6 ± 0.7 | 0.006 |
| Heart rate (per min) | 139.8 ± 1.8 | 143.3 ± 2.3 | 0.231 |
| Heart sound (abnormal) | 9 (69.2) | 4 (30.8) | 0.508 |
| Blood examination | | | |
| Hemoglobin (g/dl) | 10.5 ± 0.2 | 10.6 ± 0.2 | 0.801 |
| Total leucocyte count (per mm$^3$) | 14094.1 ± 1005.9 | 11735.3 ± 730.0 | 0.049 |
| Absolute neutrophil count (per mm$^3$) | - | - | - |
| Platelets (per mm$^3$) | 333664.9 ± 1689.1 | 303379.3 ± 16429.8 | 0.116 |
| Complications (Yes) | | | |
| No | 74 (58.3) | 53 (41.7) | 0.002 |
| Yes | 22 (91.7) | 2 (8.3) | |
| Empyema | 9 (100.0) | 0 | 0.441 |
| Myocarditis | 4 (80.0) | 1 (20.0) | |
| Parapneumonic effusion | 4 (100.0) | 0 | |
| Respiratory failure | 3 (100.0) | 0 | |
| Septic shock | 2 (66.7) | 1 (33.3) | |

(96.8%), bronchial breathing (100%) and decreased breath sounds (92.1%) had the highest specificity (Table 2).

Table 3 shows the validity of a combination of clinical variables for the prediction of pneumonia. Tachypnea alone has a high sensitivity but poor specificity (6%). The addition of hypoxia increased its specificity to 59% while further addition of various auscultatory findings (crepitations, bronchial breathing sounds, decreased air entry) increased specificity to 100%.

Univariate regression analysis, it was found that noisy breathing (OR 0.34; 95% CI 0.17–0.67), nasal discharge (OR 0.42; 95% CI 0.20–0.91), wheezing (OR 0.29; 95% CI 0.15–0.57), decreased breath sounds (OR 7.15; 95% CI 2.63–19.46), and hypoxemia (OR 2.54; 95% CI 1.32–4.88) were significantly associated with radiological pneumonia (Table 4). Following

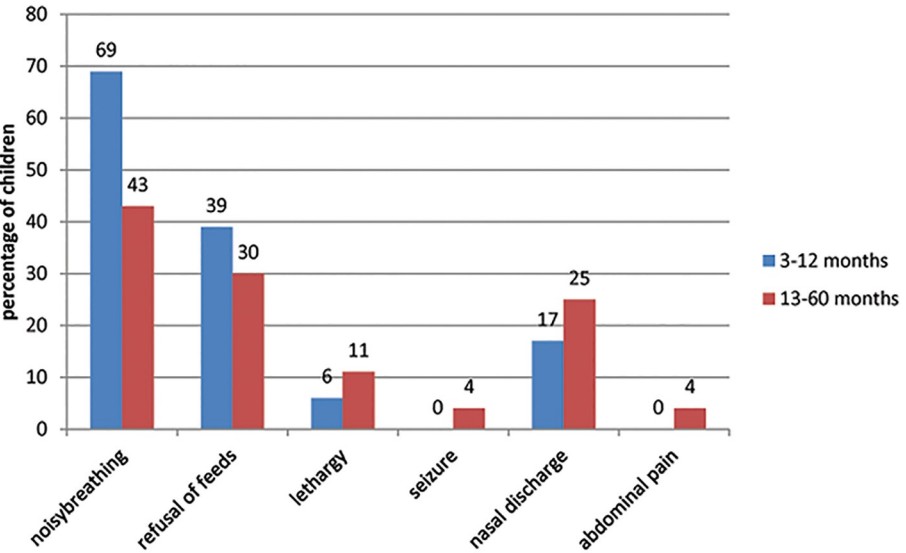

**Fig 2. Common presenting symptoms of enrolled children.**

adjustment for all the clinical signs and symptoms, only hypoxemia (AOR 3.41; 95% 95% CI 1.47–7.92) was independently associated with radiological pneumonia (Table 4).

## Discussion

In our study, tachypnea had high sensitivity and poor specificity for the diagnosis of radiological pneumonia. Although radiography is the gold standard in the diagnosis of pneumonia in low-income countries, including in Nepal, the unavailability of x-ray machines in majority of rural health settings poses a diagnostic challenge. Equally, it is not feasible to undergo a chest x-ray examination in all children with cough due to its high frequency and radiation hazards. We, therefore, still rely on simple clinical signs as laid out by WHO for diagnosing and treating

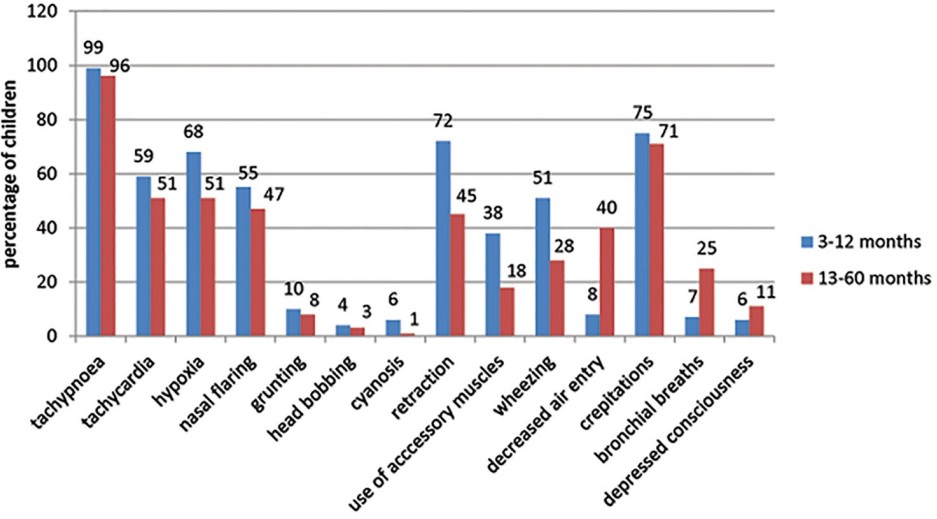

**Fig 3. Commonly observed signs in enrolled children.**

Table 2. Comparison of clinical features between children with and without radiological pneumonia.

| Clinical features | Radiological pneumonia | | | Sensitivity [95% CI] | Specificity [95% CI] |
|---|---|---|---|---|---|
| | Yes (N, %) | No (N, %) | p-value* | | |
| **Symptoms** | | | | | |
| Noisy breathing | 43 (49.4) | 44 (50.6) | 0.002 | 43/97 (44.3) [34.2–54.8] | 19/63 (30.2) [19.2–43.0] |
| Refusal of feeds | 33 (60.0) | 22 (40.0) | 0.907 | 33/97 (34.0) [24.7–44.3] | 41/63 (65.1) [52.0–76.7] |
| Lethargy | 8 (57.1) | 6 (42.9) | 0.78 | 8/97 (8.3) [3.6–15.6] | 57/63 (90.5) [80.4–96.4] |
| Nasal discharge | 15 (44.1) | 19 (55.9) | 0.026 | 15/97 (15.5) [8.9–24.2] | 44/63 (69.8) [57.0–80.8] |
| **Signs** | | | | | |
| Tachypnea | 96 (61.9) | 59 (38.1) | 0.059 | 96/97 (99.0) [94.4–100.0] | 4/63 (6.35) [1.76–15.5] |
| Nasal flaring | 54 (66.7) | 27 (33.3) | 0.113 | 54/97 (55.7) [45.2–65.8] | 36/63 (57.1) [44.0–69.5] |
| Grunting | 12 (85.7) | 2 (14.3) | 0.044 | 12/97 (12.4) [6.6–20.6] | 61/63 (96.8) [89.0–99.6] |
| Hypoxemia | 65 (69.9) | 28 (30.1) | 0.005 | 65/97 (67) [56.7–76.2] | 35/63 (55.6) [42.5–68.1] |
| Retraction | 60 (65.9) | 31 (34.1) | 0.114 | 60/97 (61.9) [51.4–71.5] | 32/63 (50.8) [37.9–63.6] |
| Wheezing | 26 (42.6) | 35 (57.4) | <0.001 | 26/97 (26.8) [18.3–36.8] | 28/63 (44.4) [31.9–57.5] |
| Decreased breath sound | 37 (88.1) | 5 (11.9) | <0.001 | 37/97 (38.1) [28.5–48.6] | 58/63 (92.1) [82.4–97.4] |
| Bronchial breath | 27 (100.0) | 0 | <0.001 | 27/97 (27.8) [19.2–37.9] | 63/63 (100.0) [94.3–100.0] |
| Crepitations | 73 (62.9) | 43 (37.1) | 0.332 | 73/97 (75.3) [65.5–83.5] | 20/63 (31.7) [20.6–44.7] |

CI, confidence interval;

*Chi square test (categorical variables).

pneumonia. WHO defines tachypnea as a sensitive sign of pneumonia; however, it has a poor specificity [13]. Hence, using only tachypnea as a guideline to define pneumonia leads to over-diagnosis of pneumonia resulting in over-prescription of antibiotics [14, 15]. Therefore, chest retraction was added in the definition of pneumonia, along with fast breathing in the WHO pocket book [3]. Chest retraction had a sensitivity of 62% in diagnosing radiological pneumonia with a specificity of 50.8% in the present study. This tachypnea based algorithms also significantly under-diagnose wheezy diseases. Likewise, specific signs like nasal flaring, retraction, hypoxemia, crepitations and wheezing may be present in asthma and cardiac diseases [15–17]. Using these specific signs may under-diagnose pneumonia cases. Therefore, a combination of clinical variables (signs and symptoms) that define pneumonia is required for its effective management [15, 18].

The prevalence of radiological pneumonia in this study was 61%, as has also been reported by earlier studies [19–21]. However, this was in contrast to other studies where the prevalence of radiological pneumonia was low [22–25]. Our study had strict inclusion criteria (cough, fever of 100.4° F or more, fast or difficulty breathing) in defining clinical pneumonia, whereas other studies used the earlier WHO definition of pneumonia (only cough and fast breathing)

Table 3. Validity of combination of variables.

| Combination of variables | Sensitivity | Specificity |
|---|---|---|
| Tachypnea + hypoxemia | 67 | 59 |
| Tachypnea + auscultatory findings | 20 | 100 |
| Hypoxemia+ wheezing | 43 | 50 |
| Hypoxemia + bronchial breath sounds | 44 | 100 |
| Wheezing+ bronchial breath sounds | 4 | 100 |
| Wheezing + hypoxemia + bronchial breath sounds | 7 | 100 |

**Table 4. Associations between clinical variables (signs and symptoms) with radiological pneumonia.**

| Variables | Crude Odds ratio (95% CI) | Adjusted Odds ratio* (95% CI) |
|---|---|---|
| Noisy breathing | 0.34 (0.17–0.67) | 0.46 (0.17–1.25) |
| Tachypnea | 6.51 (0.71–59.64) | - |
| Nasal discharge | 0.42 (0.20–0.91) | 0.85 (0.30–1.98) |
| Grunting | 4.30 (0.93–19.94) | 1.94 (0.26–14.46) |
| Wheezing | 0.29 (0.15–0.57) | 0.72 (0.28–1.87) |
| Decreased breath sounds | 7.15 (2.63–19.46) | 3.68 (0.99–13.76) |
| Hypoxemia | 2.54 (1.32–4.88) | **3.41 (1.47–7.92)** |
| Refusal to feed | 0.96 (0.49–1.87) | 0.77 (0.30–1.98) |
| Lethargy | 0.85 (0.28–2.59) | 0.15 (0.02–1.01) |
| Nasal flaring | 1.67 (0.88–3.17) | 1.72 (0.74–3.99) |
| Retraction | 1.67 (0.88–3.18) | 2.15 (0.87–5.31) |
| Crepitations | 1.41 (0.70–2.86) | 2.98 (0.06–8.12) |

CI, confidence interval;

*Each clinical variables were mutually adjusted for each other.

as their entry criteria. This might be the reason for the high prevalence of radiological pneumonia in the present study.

Although tachypnea, in the current study, was found to be the most sensitive sign to define pneumonia, its specificity was low, and the predictability of radiological pneumonia was insignificant. Similarly, in a study by Lynch *et al.* (2004) [21] and others [22–24], tachypnea had a sensitivity of above 95%, but was unable to distinguish children with and without radiological pneumonia. Likewise, Palafox *et al.* (2000) found that tachypnea had a sensitivity of 74% and concluded that tachypnea might be used as a useful screening clinical sign for identifying pneumonia in children [25].

Among various signs, the specificity of bronchial breath sound was 100% in diagnosing pneumonia in the present study. Similarly, grunting and decreased breath sounds had excellent specificities of 96% and 92%, respectively. This was similar to the study by Lozano *et al.* (1994), where decreased breath sounds had a specificity of 97% [18]. Lynch *et al.* (2004) concluded grunting had 100% specificity [21].

Although noisy breathing, nasal discharge, wheezing, decreased breath sounds, and hypoxemia were significantly associated with radiological pneumonia on univariate analysis, only hypoxemia was found to be independently associated with radiological pneumonia following adjustment of all the clinical signs and symptoms.

Using hypoxemia as a clinical sign had higher sensitivity (67%) with the specificity of 55.6% in predicting radiological pneumonia in the present study. In the present study, the clinical variable (hypoxemia) significantly associated with radiological pneumonia was similar to those reported by Lynch *et al.* (2004) [21] and Bilkis *et al.* [26]. The previous study conducted in the higher altitude of Nepal by Basnet *et al.* (2006–2008) found hypoxia in the majority proportion of children (62%) with pneumonia and predicted it as a sign of treatment failure and admission duration [27].

The sensitivity and specificity of chest retraction in predicting radiological pneumonia in the present study was about 62% and 51%, respectively with no significance in differentiating it from non-radiological pneumonia (p = 0.114). Hence, chest indrawing is probably an early indicator of respiratory distress that could be due to different disorders like pneumonia and

bronchiolitis. Although using chest indrawing only as a sole clinical sign is insufficient for a diagnosis of radiological pneumonia, it might still be useful to recognise children with a high risk of hypoxemia and would benefit from oxygen therapy rather than the provision of antibiotics [28].

No single clinical signs have been able to truly predict radiological pneumonia the revised WHO definition of pneumonia suggests tachypnea and/or retractions be used widely in the resource-poor settings to identify children with pneumonia. In the present study, tachypnea had high sensitivity but poor specificity, and its association with radiological pneumonia (p = 0.079) was statistically insignificant. Similar results were found in a study done by Lozano *et al.* (1994), where the specificity was low (20%) when tachypnea alone was used to diagnose radiological pneumonia. Wingerter *et al.* (2012) applied the WHO criteria to an urban population visiting the emergency department and found that only 111 met the WHO case definition of pneumonia out of 324 children diagnosed with radiological pneumonia (sensitivity 34.3%, 95% confidence interval: 29.1–39.7) suggesting that WHO criteria was neither sensitive nor specific in predicting pneumonia in younger children [29]. On a combination of clinical signs (tachypnea + auscultatory findings; hypoxemia + bronchial breath sounds) (Table 4), the specificity of predicting radiological pneumonia was 100% in the present study. Rothrack *et al.* suggested that the absence of each of the four signs (respiratory distress, tachypnea, rales, and decreased breath sounds) excludes the diagnosis of pneumonia in children [30]. Therefore, due care needs to be taken while ordering a chest x-ray or prescribing antibiotics to any children presenting with tachypnea alone.

The current study has a few limitations. First, as the present study included children up to 5 years with pneumonia, this result may not be valid for children above five years of age; however, excluding children above five years of age would not take into account the changing epidemiology and the clinical presentation of pneumonia in this age group. Second, this study did not attempt to search the etiological agents. Therefore, our study is not in a position to ascertain with a greater degree of certainty whether the change in epidemiological pattern and variation of clinical presentation of radiological pneumonia is bacterial or viral agents. Thirdly, as this study was conducted in a tertiary care hospital (respiratory physician and radiologist interpreted the data), it may be a challenge to apply these findings in the community setting where these facilities are lacking.

## Conclusion

Hypoxemia was the only independent predictor for radiological pneumonia. Tachypnea was the most sensitive sign, whereas bronchial breathing was the most specific sign of radiological pneumonia in the present study. This changing pattern in the clinical presentation and epidemiology of pediatric pneumonia could be due to the introduction of new vaccines which requires a reassessment of clinical predictors of pediatric pneumonia. A larger multi-centric study along with etiological diagnosis is necessary to re-define this changing clinical pattern of pediatric pneumonia to formulate new diagnostic guidelines and empirical antibiotics. The clinician should not rely only on a single sign or symptom and should consider a combination of clinical variables before diagnosing and treating pneumonia in children.

## Supporting information

**S1 Checklist. STROBE checklist.**
(DOC)

## Acknowledgments

The authors would like to acknowledge the administration and staffs of Tribhuvan University Teaching Hospital, Kathmandu, Nepal for their cooperation during the study.

## Author Contributions

**Conceptualization:** Sandeep Shrestha, Arun Sharma, Laxman Shrestha.

**Data curation:** Sandeep Shrestha, Nagendra Chaudhary, Laxman Shrestha, Om P. Kurmi.

**Formal analysis:** Sandeep Shrestha, Nagendra Chaudhary, Saneep Shrestha, Santosh Pathak, Arun Sharma, Laxman Shrestha, Om P. Kurmi.

**Funding acquisition:** Sandeep Shrestha.

**Investigation:** Sandeep Shrestha, Arun Sharma.

**Methodology:** Sandeep Shrestha, Nagendra Chaudhary, Santosh Pathak, Arun Sharma, Laxman Shrestha.

**Project administration:** Sandeep Shrestha, Saneep Shrestha, Arun Sharma, Laxman Shrestha.

**Resources:** Sandeep Shrestha, Santosh Pathak, Arun Sharma, Laxman Shrestha.

**Software:** Sandeep Shrestha, Nagendra Chaudhary, Saneep Shrestha, Santosh Pathak, Arun Sharma, Laxman Shrestha, Om P. Kurmi.

**Supervision:** Arun Sharma, Laxman Shrestha.

**Validation:** Sandeep Shrestha, Nagendra Chaudhary, Saneep Shrestha, Santosh Pathak, Arun Sharma, Laxman Shrestha, Om P. Kurmi.

**Visualization:** Sandeep Shrestha, Nagendra Chaudhary, Saneep Shrestha, Santosh Pathak, Laxman Shrestha, Om P. Kurmi.

**Writing – original draft:** Sandeep Shrestha, Nagendra Chaudhary, Santosh Pathak, Om P. Kurmi.

**Writing – review & editing:** Sandeep Shrestha, Nagendra Chaudhary, Saneep Shrestha, Santosh Pathak, Arun Sharma, Laxman Shrestha, Om P. Kurmi.

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
