## [Editor Report · Decision Letter 0]

10 Oct 2019

PONE-D-19-27004

Clinical predictors of radiological pneumonia in the post vaccine era: a cross sectional study from a tertiary hospital in Nepal

PLOS ONE

Dear Dr. Chaudhary,

Thank you for submitting your manuscript to PLOS ONE. After careful consideration, we feel that it has merit but does not fully meet PLOS ONE’s publication criteria as it currently stands. Therefore, we invite you to submit a revised version of the manuscript that addresses the points raised during the review process.

Please address the following comments before sending out for review:

Abstract conclusion, tachypnea is not specific at all (Table 2).The main purpose of the study is to reassess the clinical predictors of radiological pneumonia. In the introduction, please describe the existing known clinical predictors and compare the results in the discussion.Table 2, please give the % of having the symptoms/signs. For example, the % for ‘noisy breathing’ should be 43/97=44.3%. This will help understanding the estimated sensitivity and specificity.Please present the results of the multivariable analysis.Please explain how combinations of signs/symptoms were determined. Why some combinations were considered (or not considered)?

We would appreciate receiving your revised manuscript by Nov 24 2019 11:59PM. To enhance the reproducibility of your results, we recommend that if applicable you deposit your laboratory protocols in protocols.io, where a protocol can be assigned its own identifier (DOI) such that it can be cited independently in the future. For instructions see: http://journals.plos.org/plosone/s/submission-guidelines#loc-laboratory-protocols

We look forward to receiving your revised manuscript.

Kind regards,

Eric HY Lau, Ph.D.

Academic Editor

PLOS ONE

Journal Requirements:

Additional Editor Comments (if provided):

Please address the following comments before sending out for review:

1. Abstract conclusion, tachypnea is not specific at all (Table 2).

2. The main purpose of the study is to reassess the clinical predictors of radiological pneumonia. In the introduction, please describe the existing known clinical predictors and compare the results in the discussion.

3. Table 2, please give the % of having the symptoms/signs. For example, the % for ‘noisy breathing’ should be 43/97=44.3%. This will help understanding the estimated sensitivity and specificity.

4. Please present the results of the multivariable analysis.

5. Please explain how combinations of signs/symptoms were determined. Why some combinations were considered (or not considered)?
---

## [Author Response · Author response to Decision Letter 0]

13 Nov 2019

To, 13.11.2019

Eric HY Lau, Ph.D.

Academic Editor

PLOS ONE

Dear Sir,

Thank you for providing us with the opportunity to address your valuable comments (Manuscript ID: PONE-D-19-27004). The comments were helpful and we have provided point by point response to the comments and made changes accordingly in the manuscript. We hope this revised version will be acceptable but please let us know if you need some further clarifications.

Comments:

1. Abstract conclusion, tachypnea is not specific at all (Table 2).

Action taken: Thank you for pointing out the mistake. We have corrected this mistake. 

2. The main purpose of the study is to reassess the clinical predictors of radiological pneumonia. In the introduction, please describe the existing known clinical predictors and compare the results in the discussion.

Action taken: We have revised the introduction section as suggested and also have compared the known predictors in the discussion section.

3. Table 2, please give the % of having the symptoms/signs. For example, the % for ‘noisy breathing’ should be 43/97=44.3%. This will help understanding the estimated sensitivity and specificity.

Action taken: We have given the percentage of sensitivity and specificity in the table as suggested. 

4. Please present the results of the multivariable analysis.

Action taken: The results of multivariate analysis have been given in table 4. 

5. Please explain how combinations of signs/symptoms were determined. Why some combinations were considered (or not considered)?

Action taken: We just wanted to see the difference of sensitivity and specificity of the individual variables and on combining it. We did not find the increase in sensitivity in detecting radiological pneumonia on adding the variables. We also have calculated the combination of other variables which are given below. 

Combination of variables Sensitivity Specificity PPV NPV

Tachypnea + hypoxemia 67 59 71 54

Tachypnea + auscultatory findings 20 100 100 45

Hypoxemia+ wheezing 43 50 48 45

Hypoxemia + bronchial breath sounds 44 100 100 58

Wheezing+ bronchial breath sounds 4 100 100 38

Wheezing + hypoxemia + bronchial breath sounds 7 100 100 58

 If you suggest, we do not have any problem in omitting this table. 

Thanking you,

Dr. Nagendra Chaudhary

---

## [Decision Letter · Decision Letter 1]

18 Dec 2019

PONE-D-19-27004R1

Clinical predictors of radiological pneumonia in the post-vaccine era: a cross-sectional study from a tertiary hospital in Nepal

PLOS ONE

Dear Dr. Chaudhary,

Thank you for submitting your manuscript to PLOS ONE. After careful consideration, we feel that it has merit but does not fully meet PLOS ONE’s publication criteria as it currently stands. Therefore, we invite you to submit a revised version of the manuscript that addresses the points raised during the review process.

The Authors are expected to address all the criticisms by all Reviewers. In particular, please describe and discuss the findings in Figure 2 and 3 (Reviewer #1 and #3), provide description of the statistical analyses for Tables 2 and 3 (Reviewer #1), and draw conclusion directly from the study results (Reviewer #1), and clarify in the methods whether bacterial or viral CAP were identified. In additional to the above comments, please address:

Table 4. The study main findings suggested that Tachypnea had the highest sensitivity, however was not included in the multivariable model. I suggest that the authors may include all variables with p<0.1 from Table 2 in the multivariable model.The authors now added Table 3 to show the results of various combinations of signs/symptoms. However, a description of how the authors come up with these combinations would be helpful.

We would appreciate receiving your revised manuscript by Feb 01 2020 11:59PM. To enhance the reproducibility of your results, we recommend that if applicable you deposit your laboratory protocols in protocols.io, where a protocol can be assigned its own identifier (DOI) such that it can be cited independently in the future. For instructions see: http://journals.plos.org/plosone/s/submission-guidelines#loc-laboratory-protocols

We look forward to receiving your revised manuscript.

Kind regards,

Eric HY Lau, Ph.D.

Academic Editor

PLOS ONE

Additional Editor Comments (if provided):

The Authors are expected to address all the criticisms by all Reviewers. In particular, please describe and discuss the findings in Figure 2 and 3 (Reviewer #1 and #3), provide description of the statistical analyses for Tables 2 and 3 (Reviewer #1), and draw conclusion directly from the study results (Reviewer #1), and clarify in the methods whether bacterial or viral CAP were identified. In additional to the above comments, please address:

1. Table 4. The study main findings suggested that Tachypnea had the highest sensitivity, however was not included in the multivariable model. I suggest that the authors may include all variables with p<0.1 from Table 2 in the multivariable model.

2. The authors now added Table 3 to show the results of various combinations of signs/symptoms. However, a description of how the authors come up with these combinations would be helpful.

Reviewers' comments:

Reviewer's Responses to Questions

**Comments to the Author**

1. If the authors have adequately addressed your comments raised in a previous round of review and you feel that this manuscript is now acceptable for publication, you may indicate that here to bypass the “Comments to the Author” section, enter your conflict of interest statement in the “Confidential to Editor” section, and submit your "Accept" recommendation.

Reviewer #1: (No Response)

Reviewer #2: (No Response)

Reviewer #3: (No Response)

2. Is the manuscript technically sound, and do the data support the conclusions?

Reviewer #1: No

Reviewer #2: Yes

Reviewer #3: Partly

3. Has the statistical analysis been performed appropriately and rigorously? 

Reviewer #1: Yes

Reviewer #2: Yes

Reviewer #3: No

4. Have the authors made all data underlying the findings in their manuscript fully available?

Reviewer #1: Yes

Reviewer #2: Yes

Reviewer #3: No

5. Is the manuscript presented in an intelligible fashion and written in standard English?

Reviewer #1: Yes

Reviewer #2: Yes

Reviewer #3: No

6. Review Comments to the Author

Reviewer #1: 1. In the manuscript, the name of the microorganism should be corrected as Haemophilus influenzae. It is the way it has been used in the bacteriology world since the organism has been defined.

2. In the Methods section, “Ethics Approval” statement should be replaced after “Study Outcomes”.

3. In the Statistical Analysis subsection, it was indicated that statistical analysis was performed using SPSS. The name of the company and country should be indicated as “IBM SPSS Statistics (version 21.0; IBM, Armonk, NY, US)”.

4. In Figure 2 ans 3, symptoms and signs of the patients were presented in an age-based grouping. However, this analysis type has not been indicated in the methods section. Additionally, in the results and discussion sections this analysis has not been commented in any way. These figures give no input into the manuscript. This ag- based analysis and Figure 2 and 3 are unnecessary in the manuscript.

5. In the results section, explanations of Table and Table 3 should be more informative. The frequencies and statistical differences should be explained in a more detailed fashion, since these data are the main ones that the manuscipt is based on

6. The statistical analysis of Tables 2,3, and 4 are not described sufficient detail.

7. In the first paragraph of Discussion section, there is no data discussed from the present manuscript. This paragraph is just explaining what is already known in this era and it is a repeat of introduction section.

8. The manuscript is presenting a cross-sectional analysis and it does not compare the pre and post-vaccine era data. Both in introduction section and discussion sections, the authors are referring to a change of epidemiology and its reflections on the radiology and clinical signs. However, with the data presented in the manuscript, these evaluations can not be performed. Although authors are hypothizing changes in clinical findings of pneumonia related to vaccine introduction, they did not analyze this in the manuscript.

9. Fourth paragraph of manuscript (line 232), the discussion of the data is weak in explanations.

10. The conclusion section, totally, is not appropriate as a conclusion of this manuscript. It is not supported by the data.

Reviewer #2: About the scientific paper: "Clinical predictors of radiological pneumonia in the post-vaccine era: a cross-sectional study from a tertiary hospital in Nepal"

-A)First, the design , exclusion and inclusion criterial is similar to another old historical cohort of clinical predictors of community acquired pneumonia.

-B)Use of radiological finding as gold standard , not only WHO clinical guide is right.

-C)"The clinical presentation and the radiographic signs of pneumonia may not be the same as found earlier. Considering the change in the epidemiological pattern, the clinical predictors of pediatric pneumonia need

reassessment. Therefore, this hospital-based study was conducted to find the predictors of radiological pneumonia in the post-vaccine era".

This statement is a hypothesis of the authors of this paper and has no bibliographic work to support it.

-D)"Currently, with the introduction of Hemophilus influenzae type b (Hib) and pneumococcal conjugate vaccines (PCV) and global expansion of their coverage,

bacterial agents are on the decline and out-numbered by viral and atypical bacteria".

In this serie you studied include infants with a single dose of conjugate vaccines, which is considered insufficient dose to ensure adequate coverage.

Data that would not emphatically exclude the possible presence of pneumococci or haemophillus as possible etiological agents.

-E)Nasal discharge is inespecific and is associated with flu, the inicial and common manifestation of respiratory symptoms, not only of pneumonia.

-F)The association of wheezing with pulmonary infiltrates in infants is the current basis for the recommendation not to do radiography in bronchiolitis,

given the frequent association of pulmonary pathological findings and possible diagnostic confusion with pneumonia.

Would this have happened in some of the patients included in this series?. The same can be said of the asthmatic crisis.

-G)The central paper hypothesis cannot be demonstrated, given that these clinical predictors are not compared with others taken in the era prevaccine in the same community and population studied.

-H)The prevalence of viruses over bacteria in the etiology of pneumonia alters the clinical presentation of pneumonia ?.

Due to the design in a single hospital and without accompanying etiological studies, it is difficult to answer this question. The works cited do not get into this disquisition.

-I)In conclusion, it is valuable to observe what the high altitude above sea level determines that hypoxemia can be a good predictor of pneumonia.

The design and casuistic are adequate and the sum of signs and symptoms help and much to be able to find and not lose cases of this disease as severe and harmful to children as pneumonia.

By making these changes, these findings could be communicated.

Reviewer #3: The authors conducted an interesting study to analyze a possible association between radiographic findings of pneumonia in children with clinical signs and symptoms. The results are equally interesting, however, some questions need clarifying or undergoing changes to make the article better quality.

a) Were the selected cases all of community-acquired pneumonia? If so, this should be made clear in the text and the expression 'pneumonia' should be replaced throughout 'community acquired pneumonia (CAP)'.

b) The authors do not clarify whether bacterial or viral CAP was identified, this is only briefly described in the discussion (in limitations). This doubt should be clarified in the methods, emphasizing that approximately 30% of the cases of CAP are mixed (viral and bacterial).

c) How was the positive predictive value (PPV) calculated? If it was calculated from the selected sample, the value is certainly overestimated. It is correct to calculate PPV based on population prevalence data using the Bayes equation.

d) An interesting analysis would be to evaluate the association/correlation between radiological findings and leukocyte count, considering that in bacterial PAC, there is a tendency for leukocytosis.

e) The analysis of age subgroups would also be very important.

f) The authors use and cite old references. I suggest the inclusion and citation of some recent references such as: DOI: 10.1016/S1473-3099(15)70017-4; DOI: 10.4046/trd.2015.78.3.196; among others.

g) All text should be revised by a native speaker of the English language and specializes in health issues.

7. PLOS authors have the option to publish the peer review history of their article (what does this mean?). If published, this will include your full peer review and any attached files.

Reviewer #1: No

Reviewer #2: Yes: Dr Manuel D. Bilkis

Reviewer #3: No

---

## [Author Response · Author response to Decision Letter 1]

28 Jan 2020

REPLY TO THE COMMENTS OF THE REVIEWERS

Additional Editor Comments (if provided):

The Authors are expected to address all the criticisms by all Reviewers. In particular, please describe and discuss the findings in Figure 2 and 3 (Reviewer #1 and #3), provide description of the statistical analyses for Tables 2 and 3 (Reviewer #1), and draw conclusion directly from the study results (Reviewer #1), and clarify in the methods whether bacterial or viral CAP were identified.

Action taken: The findings of figures 2 and 3 have been described and discussed in the results section. 

In table 2, chi- square test was used as the variable were categorical and finally sensitivity, specificity, PPV and NPV were calculated using the formula (by creating 2X2 table).

Clinical variable Radiological Pneumonia Radiological Pneumonia

 Yes No

Yes True Positive (TP) False Positive (FP)

No False Negative (FN) True negative (TN)

Sensitivity=TP/(TP+FN)

Specificity= TN/(TN+FP)

PPV=TP/(TP+FP)

NPV=TN/(TN+FN)

Clarification of the viral/bacterial CAP have been mentioned in the Methods

(In section: Study design, hospital setting, participants and diagnosis). 

In additional to the above comments, please address:

1. Table 4. The study main findings suggested that Tachypnea had the highest sensitivity, however was not included in the multivariable model. I suggest that the authors may include all variables with p<0.1 from Table 2 in the multivariable model.

Action taken: We have re-analysed the data to provide univariate and multivariable adjustment for Table 4 with all the clinical variables from Table 2 in the model except bronchial breath.. The model for Tachypnea did not converge following multiple adjustment and hence represented by dash. This could be due to small sample size.

2. The authors now added Table 3 to show the results of various combinations of signs/symptoms. However, a description of how the author come up with these combinations would be helpful.

Action taken: The exact diagnosis of pneumonia depends on various signs and symptoms. As you are aware that no single sign or symptom is sensitive or specific for the diagnosis of pneumonia. Therefore, practically clinician combines different signs and symptoms for the final diagnosis of pneumonia. We randomly combined 2 or 3 variables to see how the sensitivity or specificity changes and tried to give a message whether the combination of those variables may be able to predict pneumonia exactly. 

Reviewers' comments:

Reviewer's Responses to Questions

Comments to the Author

1. If the authors have adequately addressed your comments raised in a previous round of review and you feel that this manuscript is now acceptable for publication, you may indicate that here to bypass the “Comments to the Author” section, enter your conflict of interest statement in the “Confidential to Editor” section, and submit your "Accept" recommendation.

Reviewer #1: (No Response)

Reviewer #2: (No Response)

Reviewer #3: (No Response)

2. Is the manuscript technically sound, and do the data support the conclusions?

Reviewer #1: No

Reviewer #2: Yes

Reviewer #3: Partly

Action taken: We have amended the manuscript based on the helpful comments from the reviewers. We believe the inferences derived in this revised version are supported by the data and is technically sound.

3. Has the statistical analysis been performed appropriately and rigorously? 

Reviewer #1: Yes

Reviewer #2: Yes

Reviewer #3: No

 Action taken: We have reviewed the statistical section and the numbers. They are correct. We have dropped Table 4 to remove the ambiguity and now the message is much simpler and clearer.

4. Have the authors made all data underlying the findings in their manuscript fully available?

Reviewer #1: Yes

Reviewer #2: Yes

Reviewer #3: No

 Action taken: We have made all the data available and have uploaded the excel sheet of the study population. 

5. Is the manuscript presented in an intelligible fashion and written in standard English?

Reviewer #1: Yes

Reviewer #2: Yes

Reviewer #3: No

 Action taken: We have checked the language and edited wherever required. Typographical or grammatical errors have been corrected as far as possible by the authors.

6. Review Comments to the Author

Reviewer #1: 

1. In the manuscript, the name of the microorganism should be corrected as Haemophilus influenzae. It is the way it has been used in the bacteriology world since the organism has been defined.

Action taken: The microorganism “Haemophilus influenzae” has been corrected as suggested. 

2. In the Methods section, “Ethics Approval” statement should be replaced after “Study Outcomes”.

Action taken: The “Ethics approval” statement has been replaced after the “Study Outcomes” as suggested. 

3. In the Statistical Analysis subsection, it was indicated that statistical analysis was performed using SPSS. The name of the company and country should be indicated as “IBM SPSS Statistics (version 21.0; IBM, Armonk, NY, US)”.

Action taken: The modification has been done in the Statistical analysis subsection. 

4. In Figure 2 and 3, symptoms and signs of the patients were presented in an age-based grouping. However, this analysis type has not been indicated in the methods section. Additionally, in the results and discussion sections this analysis has not been commented in any way. These figures give no input into the manuscript. This age- based analysis and Figure 2 and 3 are unnecessary in the manuscript.

Action taken: We thank you for your recommendation and suggestion. In figure 2 and 3, the various clinical signs and symptoms have been mentioned. We have added the details in the method section as advised by you. We feel that the figures 2 and 3 depict the frequency of various clinical signs/symptoms in ≤12 months and ≥13 months children and will help in understanding the clinical signs/symptoms in children with pneumonia. The clinical predictors also have been addressed in the discussion section as advised. 

5. In the results section, explanations of Table and Table 3 should be more informative. The frequencies and statistical differences should be explained in a more detailed fashion, since these data are the main ones that the manuscript is based on.

Action taken: The frequencies and explanations of tables have been elaborated as much as possible. We feel the tables are self-explanatory and adding too much of details from the tables would be repetition of the results. However, we have added the main result findings in the result section as suggested. 

6. The statistical analysis of Tables 2, 3, and 4 are not described sufficient detail.

Action taken: The statistical analysis was done in SPSS software by creating two categorical variables (radiological pneumonia versus non-radiological pneumonia) and chi-square test was done to find the level of significance. This has been mentioned in the material and methods (statistical analysis) section. 

Again the sensitivity, specificity, PPV and NPV was calculated by using the formula mentioned above. 

7. In the first paragraph of Discussion section, there is no data discussed from the present manuscript. This paragraph is just explaining what is already known in this era and it is a repeat of introduction section.

Action taken: We have modified the first paragraph of the discussion section and reported the main findings from this study 

8. The manuscript is presenting a cross-sectional analysis and it does not compare the pre and post-vaccine era data. Both in introduction section and discussion sections, the authors are referring to a change of epidemiology and its reflections on the radiology and clinical signs. However, with the data presented in the manuscript, these evaluations cannot be performed. Although authors are hypothizing changes in clinical findings of pneumonia related to vaccine introduction, they did not analyze this in the manuscript.

Action taken: We thank you for your valuable comments. The main objective of the present study was to reassess the clinical predictors of radiological pneumonia in the post vaccine era. We compared the clinical predictors used to diagnose pneumonia in the past (pre-vaccine era) but we agree this might have created confusion. We have therefore modified the title and also reference to it in the manuscript.

9. Fourth paragraph of manuscript (line 232), the discussion of the data is weak in explanations.

Action taken: Thank you for your valuable suggestions. We have elaborated the discussion in this paragraph as suggested by you. 

10. The conclusion section, totally, is not appropriate as a conclusion of this manuscript. It is not supported by the data.

Action taken: We have modified the conclusion section which now supports the data. 

Reviewer #2: About the scientific paper: "Clinical predictors of radiological pneumonia in the post-vaccine era: a cross-sectional study from a tertiary hospital in Nepal"

-A) First, the design, exclusion and inclusion criteria is similar to another old historical cohort of clinical predictors of community acquired pneumonia.

Action taken: This study is a cross-sectional study where we have tried to reassess the clinical predictors of radiological pneumonia. Previous old studies were done in the pre-immunization era. Therefore our main objective was to reassess the clinical signs/symptoms of radiological pneumonia. 

-B) Use of radiological finding as gold standard, not only WHO clinical guide is right.

Action taken: We have therefore classified clinical pneumonia (as per WHO pocket book) into 2 groups (radiological pneumonia and non-radiological pneumonia) and compared different clinical variables. 

-C)"The clinical presentation and the radiographic signs of pneumonia may not be the same as found earlier. Considering the change in the epidemiological pattern, the clinical predictors of pediatric pneumonia need reassessment. Therefore, this hospital-based study was conducted to find the predictors of radiological pneumonia in the post-vaccine era".

This statement is a hypothesis of the authors of this paper and has no bibliographic work to support it.

Action taken: Thank you for your valuable comments. Main objective of our study was to compare the clinical signs/symptoms of radiological pneumonia with non-radiological pneumonia. We have compared this findings with previous studies in the discussion section. 

As suggested by you, we have removed the phrase “Post vaccine era” from title and other places in reassessing the clinical predictors of radiological pneumonia. 

-D)"Currently, with the introduction of Hemophilus influenzae type b (Hib) and pneumococcal conjugate vaccines (PCV) and global expansion of their coverage,

bacterial agents are on the decline and out-numbered by viral and atypical bacteria".

In this series you studied include infants with a single dose of conjugate vaccines, which is considered insufficient dose to ensure adequate coverage.

Data that would not emphatically exclude the possible presence of pneumococci or Haemophilus as possible etiological agents.

Action taken: Children aged 3-60 months were included in the present study. The Hib vaccine in Nepal is given at 6 weeks, 10 weeks and 14 weeks whereas PCV is given at 6 weeks, 10 weeks and 9 months as per the national immunization schedule. We therefore, made sure that the infant should at least receive the first dose of hib and PCV for enrolment in this study. This has been mentioned in the material and methods section. 

We were unable to perform the etiological diagnosis for pneumonia in this study which we have mentioned in the limitation section. We, therefore, accept your point and consider to conduct future study as commented by you. 

-E)Nasal discharge is inespecific and is associated with flu, the inicial and common manifestation of respiratory symptoms, not only of pneumonia.

Action taken: The sensitivity of nasal discharge in our study too was just 15% in predicting radiological pneumonia. The comparison, sensitivity and specificity has been mentioned in table 2.

F) The association of wheezing with pulmonary infiltrates in infants is the current basis for the recommendation not to do radiography in bronchiolitis, given the frequent association of pulmonary pathological findings and possible diagnostic confusion with pneumonia.

Would this have happened in some of the patients included in this series?. The same can be said of the asthmatic crisis.

Action taken: We agree that the diagnosis of pneumonia should include not only a single sign or symptoms but combination of them (as per WHO) along with radiology. In the present study, radiological diagnosis of pneumonia was made in children with clinical pneumonia by an experienced radiologist in a blinded condition to exclude observer bias. Therefore, as radiological pneumonia diagnosis was standardized, it is less likely that other diagnosis apart from pneumonia (bronchiolitis, asthma) were included. Still, we accept that there could be a possibility as suspected by you as we were unable to exclude bronchiolitis by determining the etiology (viral cause). 

G) The central paper hypothesis cannot be demonstrated, given that these clinical predictors are not compared with others taken in the era pre-vaccine in the same community and population studied.

Action taken: We have compared the predictors of pneumonia in the past (outside Nepal) with the present study in the discussion section. We could not find any studies conducted in Nepal in the pre-vaccine era in the same community except a study conducted by Basnet et al in 2006-2008. We, therefore, in this study, have tried to reassess the clinical predictors of radiological pneumonia by comparing studies conducted on predictors of pneumonia in the past. 

We, however, have tried to remove the pre and post vaccine era in the discussion wherever possible as suggested by you. 

H) The prevalence of viruses over bacteria in the etiology of pneumonia alters the clinical presentation of pneumonia?

Due to the design in a single hospital and without accompanying etiological studies, it is difficult to answer this question. The works cited do not get into this disquisition.

Action taken: We have already mentioned that the etiological diagnosis could not be done in the limitation section. This could be a future research to clinically predict pneumonia with etiology. 

It is already known that bacterial causes of pneumonia is on declining trend due to universal coverage of PCV and Hib vaccines in the immunisation schedule worldwide. Our findings of predictors of pneumonia has been conducted in children who have been vaccinated with Hib and PCV vaccines. Even though we were unable to perform the etiological diagnosis (mentioned as one of the limitation of the study), it is well understood that these cases of pneumonia could have been due to viral cause in majority. Therefore, we feel that this statement is justified although we accept that etiological diagnosis is required for confirmation. 

I) In conclusion, it is valuable to observe what the high altitude above sea level determines that hypoxemia can be a good predictor of pneumonia. The design and casuistic are adequate and the sum of signs and symptoms help and much to be able to find and not lose cases of this disease as severe and harmful to children as pneumonia.

By making these changes, these findings could be communicated.

Action: Hypoxia is an important predictor of radiological pneumonia. This study was conducted at 1400 meters (4600 ft) from the sea level. Previous study was also conducted at the same altitude of the nation and found that hypoxia was a predictor of duration of admission and treatment failure.

We have added in the discussion section. 

Reviewer #3: The authors conducted an interesting study to analyze a possible association between radiographic findings of pneumonia in children with clinical signs and symptoms. The results are equally interesting, however, some questions need clarifying or undergoing changes to make the article better quality.

a) Were the selected cases all of community-acquired pneumonia? If so, this should be made clear in the text and the expression 'pneumonia' should be replaced throughout 'community acquired pneumonia (CAP)'.

Action taken: All the cases were community acquired pneumonia. It has been added in the material and methods (Study design, hospital setting, participants and diagnosis). 

b) The authors do not clarify whether bacterial or viral CAP was identified, this is only briefly described in the discussion (in limitations). This doubt should be clarified in the methods, emphasizing that approximately 30% of the cases of CAP are mixed (viral and bacterial).

Action taken: As etiological diagnosis was not done in the study, we are unable to classify the pneumonia cases etiologically (viral or bacterial). We already have mentioned it in the limitation section.

As suggested by you, the same has been added in the methods section. 

c) How was the positive predictive value (PPV) calculated? If it was calculated from the selected sample, the value is certainly overestimated. It is correct to calculate PPV based on population prevalence data using the Bayes equation.

Action: We calculated PPV and NPV using the following method. 

Clinical variable Radiological Pneumonia Yes Radiological Pneumonia

No

Yes True Positive (TP) False Positive (FP)

No False Negative (FN) True negative (TN)

Sensitivity=TP/(TP+FN)

Specificity= TN/(TN+FP)

PPV=TP/(TP+FP)

NPV=TN/(TN+FN)

d) An interesting analysis would be to evaluate the association/correlation between radiological findings and leukocyte count, considering that in bacterial PAC, there is a tendency for leucocytosis.

Action taken: The association of TLC with radiological pneumonia has been given in table 1. We have mentioned it on the results section too. We found significantly high TLC values in radiological pneumonia when compared to non-radiological pneumonia. 

e) The analysis of age subgroups would also be very important.

Action taken: We accept your suggestion that the age sub analysis would be important, however, this will mean the sample size in each age category will be too small and hence the result will likely to be biased. A much larger study would be needed to carry out sub-group analysis for age.

f) The authors use and cite old references. I suggest the inclusion and citation of some recent references such as: DOI: 10.1016/S1473-3099(15)70017-4; DOI: 10.4046/trd.2015.78.3.196; among others.

Action taken: We have updated more recent references and included the references as suggested by you. 

g) All text should be revised by a native speaker of the English language and specializes in health issues.

Action taken: Thank you for the suggestion. The manuscript has been rechecked for grammatical errors.

---

## [Decision Letter · Decision Letter 2]

21 Apr 2020

PONE-D-19-27004R2

Clinical predictors of radiological pneumonia: a cross-sectional study from a tertiary hospital in Nepal

PLOS ONE

Dear Dr. Chaudhary,

Thank you for submitting your manuscript to PLOS ONE. After careful consideration, we feel that it has merit but does not fully meet PLOS ONE’s publication criteria as it currently stands. Therefore, we invite you to submit a revised version of the manuscript that addresses the points raised during the review process.

We would appreciate receiving your revised manuscript. To enhance the reproducibility of your results, we recommend that if applicable you deposit your laboratory protocols in protocols.io, where a protocol can be assigned its own identifier (DOI) such that it can be cited independently in the future. For instructions see: http://journals.plos.org/plosone/s/submission-guidelines#loc-laboratory-protocols

We look forward to receiving your revised manuscript.

Kind regards,

Frederick Quinn

Academic Editor

PLOS ONE

Reviewers' comments:

Reviewer's Responses to Questions

**Comments to the Author**

1. If the authors have adequately addressed your comments raised in a previous round of review and you feel that this manuscript is now acceptable for publication, you may indicate that here to bypass the “Comments to the Author” section, enter your conflict of interest statement in the “Confidential to Editor” section, and submit your "Accept" recommendation.

Reviewer #1: (No Response)

Reviewer #2: All comments have been addressed

Reviewer #3: (No Response)

2. Is the manuscript technically sound, and do the data support the conclusions?

Reviewer #1: Partly

Reviewer #2: Yes

Reviewer #3: Partly

3. Has the statistical analysis been performed appropriately and rigorously? 

Reviewer #1: I Don't Know

Reviewer #2: Yes

Reviewer #3: No

4. Have the authors made all data underlying the findings in their manuscript fully available?

Reviewer #1: Yes

Reviewer #2: Yes

Reviewer #3: Yes

5. Is the manuscript presented in an intelligible fashion and written in standard English?

Reviewer #1: No

Reviewer #2: Yes

Reviewer #3: (No Response)

6. Review Comments to the Author

Reviewer #1: 1) Although most of the reviewer comments were met by the authors in the revision process, I still think that the manuscript does not add any new data into both clinical practice and the literature for further studies.

2) Since the statistical data analysis has already been explained in Results section, the numbers should not be repeated in Discussion section (all the confidence intervals, percentages and so forth).

Reviewer #2: (No Response)

Reviewer #3: The calculation of the positive predictive value (PPV) should be done using the Bayes equation, using the prevalence of real pneumonia in the child population in Nepal and not using data from the 2x2 table (see: Linn S. New patient-oriented diagnosis test characteristics analogous to the likelihood ratios conveyed information on trustworthiness. J Clin Epidemiol. 2005; 58 (5): 450–457. doi: 10.1016 / j.jclinepi.2004.07.009 - see equation 2.11).

If this cannot be done by the authors, I suggest not presenting these values in the manuscript.

7. PLOS authors have the option to publish the peer review history of their article (what does this mean?). If published, this will include your full peer review and any attached files.

Reviewer #1: No

Reviewer #2: No

Reviewer #3: Yes: Altacilio Nunes

---

## [Author Response · Author response to Decision Letter 2]

29 Apr 2020

REPLY TO THE COMMENTS OF THE REVIEWERS

Comments to the Author

1. If the authors have adequately addressed your comments raised in a previous round of review and you feel that this manuscript is now acceptable for publication, you may indicate that here to bypass the “Comments to the Author” section, enter your conflict of interest statement in the “Confidential to Editor” section, and submit your "Accept" recommendation.

Reviewer #1: (No Response)

Reviewer #2: All comments have been addressed

Reviewer #3: (No Response________________________________________

2. Is the manuscript technically sound, and do the data support the conclusions?

Reviewer #1: Partly

Reviewer #2: Yes

Reviewer #3: Partly

Action taken: We have again amended the manuscript based on the helpful comments from the reviewers. We believe the inferences derived in this revised version are supported by the data and is technically sound.________________________________________

3. Has the statistical analysis been performed appropriately and rigorously? 

Reviewer #1: I Don't Know

Reviewer #2: Yes

Reviewer #3: No

Action taken: We have again reviewed the statistical section and the numbers. They are correct.________________________________________

4. Have the authors made all data underlying the findings in their manuscript fully available?

Reviewer #1: Yes

Reviewer #2: Yes

Reviewer #3: Yes________________________________________

5. Is the manuscript presented in an intelligible fashion and written in Standard English?

Reviewer #1: No

Reviewer #2: Yes

Reviewer #3: (No Response)

Action taken: We have once again checked the language and edited wherever required. Typographical or grammatical errors have been corrected as far as possible by the authors.________________________________________

6. Review Comments to the Author

Reviewer #1: 1) Although most of the reviewer comments were met by the authors in the revision process, I still think that the manuscript does not add any new data into both clinical practice and the literature for further studies.

Action taken: We have mentioned the aim and objective of the study in the manuscript. The diagnosis of pneumonia in children remains an important yet difficult clinical problem. The clinical presentation and the radiographic signs of pneumonia may not be the same as found earlier. Considering the change in the epidemiological pattern, the clinical predictors of pediatric pneumonia need reassessment. Hence, this hospital-based study was conducted to find the predictors of radiological pneumonia. We therefore, feel that this study definitely adds new data to the changing epidemiology of pneumonia. 

2) Since the statistical data analysis has already been explained in Results section, the numbers should not be repeated in Discussion section (all the confidence intervals, percentages and so forth).

Action taken: Thank you for your suggestion. We have removed the repetition and amended the manuscript as advised. 

Reviewer #2: (No Response)

Reviewer #3: The calculation of the positive predictive value (PPV) should be done using the Bayes equation, using the prevalence of real pneumonia in the child population in Nepal and not using data from the 2x2 table (see: Linn S. New patient-oriented diagnosis test characteristics analogous to the likelihood ratios conveyed information on trustworthiness. J Clin Epidemiol. 2005; 58 (5): 450–457. doi: 10.1016 / j.jclinepi.2004.07.009 - see equation 2.11).

If this cannot be done by the authors, I suggest not presenting these values in the manuscript.

Action taken: We agree to your suggestion sir. The study was conducted in a tertiary care hospital of Nepal where patients visit from different demographic locations. The data on prevalence of pneumonia in all of the population/ demo-graphical locations of Nepal are scarce. So, we could not apply the Bayes equation to calculate the PPV. We, therefore, as per your suggestion, have removed the PPV and NPV values from the manuscript.

---

## [Decision Letter · Decision Letter 3]

13 May 2020

PONE-D-19-27004R3

Clinical predictors of radiological pneumonia: a cross-sectional study from a tertiary hospital in Nepal

PLOS ONE

Dear Dr. Chaudhary

Thank you for submitting your manuscript to PLOS ONE. After careful consideration, we feel that it has merit but does not fully meet PLOS ONE’s publication criteria as it currently stands. Therefore, we invite you to submit a revised version of the manuscript that addresses the points raised during the review process.

We would appreciate receiving your revised manuscript. To enhance the reproducibility of your results, we recommend that if applicable you deposit your laboratory protocols in protocols.io, where a protocol can be assigned its own identifier (DOI) such that it can be cited independently in the future. For instructions see: http://journals.plos.org/plosone/s/submission-guidelines#loc-laboratory-protocols

We look forward to receiving your revised manuscript.

Kind regards,

Frederick Quinn

Academic Editor

PLOS ONE

Reviewers' comments:

Reviewer's Responses to Questions

**Comments to the Author**

1. If the authors have adequately addressed your comments raised in a previous round of review and you feel that this manuscript is now acceptable for publication, you may indicate that here to bypass the “Comments to the Author” section, enter your conflict of interest statement in the “Confidential to Editor” section, and submit your "Accept" recommendation.

Reviewer #1: (No Response)

Reviewer #2: All comments have been addressed

Reviewer #3: All comments have been addressed

2. Is the manuscript technically sound, and do the data support the conclusions?

Reviewer #1: Partly

Reviewer #2: Partly

Reviewer #3: Yes

3. Has the statistical analysis been performed appropriately and rigorously? 

Reviewer #1: Yes

Reviewer #2: Yes

Reviewer #3: Yes

4. Have the authors made all data underlying the findings in their manuscript fully available?

Reviewer #1: Yes

Reviewer #2: Yes

Reviewer #3: Yes

5. Is the manuscript presented in an intelligible fashion and written in standard English?

Reviewer #1: No

Reviewer #2: Yes

Reviewer #3: Yes

6. Review Comments to the Author

Reviewer #1: I still think that the manuscript does not add any new data into the literature and to the clinical practice, either. Diagnosis of pneumonia in children is not an argumentative issue with the defined clinical findings. High respiratory rate, grunting, hypoxemia, decreased breath sounds are already in practice and in literature as signs of pneumonia. Further more, the authors did not study on relation of changes in epidemiology and clinical findings. So, they can not make any comment on that. Epidemiology is the key point in giving decisions on treatment strategies, not in diagnosis of pneumonia, whether it is radiologically approved or not.

Reviewer #2: Knowing exactly the symptoms of pneumonia in children is difficult, but it is essential in developing countries.

At the time of respiratory virus pandemic, knowing the local clinical characteristics of presentation of pneumonia becomes even more important.

Reviewer #3: Decreased breathing (AOR 3.68; CI 0.99-13.76) was not independently associated, as the lower limit of 95%CI is less than 1. Please correct this, moreover, in all confidence intervals the correct form is 95%CI.

7. PLOS authors have the option to publish the peer review history of their article (what does this mean?). If published, this will include your full peer review and any attached files.

Reviewer #1: No

Reviewer #2: Yes: manuel david bilkis

Reviewer #3: Yes: Altacilio Nunes

---

## [Author Response · Author response to Decision Letter 3]

14 May 2020

REPLY TO THE COMMENTS OF THE REVIEWERS

Reviewers' comments:

Reviewer's Responses to Questions

Comments to the Author

1. If the authors have adequately addressed your comments raised in a previous round of review and you feel that this manuscript is now acceptable for publication, you may indicate that here to bypass the “Comments to the Author” section, enter your conflict of interest statement in the “Confidential to Editor” section, and submit your "Accept" recommendation.

Reviewer #1: (No Response)

Reviewer #2: All comments have been addressed

Reviewer #3: All comments have been addressed

2. Is the manuscript technically sound, and do the data support the conclusions?

Reviewer #1: Partly

Reviewer #2: Partly

Reviewer #3: Yes________________________________________

3. Has the statistical analysis been performed appropriately and rigorously? 

Reviewer #1: Yes

Reviewer #2: Yes

Reviewer #3: Yes________________________________________

4. Have the authors made all data underlying the findings in their manuscript fully available?

Reviewer #1: Yes

Reviewer #2: Yes

Reviewer #3: Yes________________________________________

5. Is the manuscript presented in an intelligible fashion and written in Standard English?

Reviewer #1: No

Reviewer #2: Yes

Reviewer #3: Yes

Action taken: We have made some amendments to the earlier submitted version of the manuscript . ________________________________________

6. Review Comments to the Author

Please use the space provided to explain your answers to the questions above. You may also include additional comments for the author, including concerns about dual publication, research ethics, or publication ethics. (Please upload your review as an attachment if it exceeds 20,000 characters).

Reviewer #1: I still think that the manuscript does not add any new data into the literature and to the clinical practice, either. Diagnosis of pneumonia in children is not an argumentative issue with the defined clinical findings. High respiratory rate, grunting, hypoxemia, decreased breath sounds are already in practice and in literature as signs of pneumonia. Furthermore, the authors did not study on relation of changes in epidemiology and clinical findings. So, they cannot make any comment on that. Epidemiology is the key point in giving decisions on treatment strategies, not in diagnosis of pneumonia, whether it is radiologically approved or not.

Action taken: We acknowledge the reviewer view point on the significance of novelty of this study, however, we differ on its importance of this study in resource scare settings of low-income countries in Nepal where there is requirement of decision making based on the clinical presentations. The diagnosis of pneumonia in children remains an important yet difficult clinical problem especially in low and middle income countries (LMICs) like Nepal. The clinical presentation and the radiographic signs of pneumonia may not be the same as found earlier. Inclusion of vaccination (Hib and PCV vaccines) have changed the epidemiology and presentation of pneumonia in children. Considering the change in the epidemiological pattern, the recent WHO pocket book has also modified the definition of pneumonia. This highlights that the clinical predictors of pediatric pneumonia need reassessment from time to time. Hence, this hospital-based study was conducted to find the predictors of radiological pneumonia in post vaccination era. We therefore, feel that this study highlights on the importance and selection of proper clinical presentation for the diagnosis of pneumonia in low-income countries. 

Reviewer #2: Knowing exactly the symptoms of pneumonia in children is difficult, but it is essential in developing countries.

At the time of respiratory virus pandemic, knowing the local clinical characteristics of presentation of pneumonia becomes even more important.

Action taken: We agree and thank you for highlighting this issue. Although the main objective of our study was also to assess the clinical predictors of radiological pneumonia in children, however, in the current pandemic, it is important to identify both typical and atypical pneumonia cases as early as possible so both preventive and therapeutic measures can be started at an early phase.

Reviewer #3: Decreased breathing (AOR 3.68; CI 0.99-13.76) was not independently associated, as the lower limit of 95%CI is less than 1. Please correct this, moreover, in all confidence intervals the correct form is 95%CI.

Action taken: Thank you. We have amended the manuscript and inserted “95%” before CI as advised. We understand the adjusted odds ratio of 3.68 with a confidence interval of 0.99 – 13.76 is statistically not significant if you use a very rigid theoretical definition, however, this can be considered as borderline significance. The wide confidence interval is most probably due to low power. We think another study with adequate sample size is required to confirm our findings. We have amended the manuscript accordingly.

---

## [Decision Letter · Decision Letter 4]

2 Jun 2020

PONE-D-19-27004R4

Clinical predictors of radiological pneumonia: a cross-sectional study from a tertiary hospital in Nepal

PLOS ONE

Dear Dr. Dr. Chaudhary,

Thank you for submitting your manuscript to PLOS ONE. After careful consideration, we feel that it has merit but does not fully meet PLOS ONE’s publication criteria as it currently stands. Therefore, we invite you to submit a revised version of the manuscript that addresses the points raised during the review process.

Please submit your revised manuscript. If you will need more time than this to complete your revisions, please reply to this message or contact the journal office at plosone@plos.org. Please include the following items when submitting your revised manuscript:

We look forward to receiving your revised manuscript.

Kind regards,

Frederick Quinn

Academic Editor

PLOS ONE

Reviewers' comments:

Reviewer's Responses to Questions

**Comments to the Author**

1. If the authors have adequately addressed your comments raised in a previous round of review and you feel that this manuscript is now acceptable for publication, you may indicate that here to bypass the “Comments to the Author” section, enter your conflict of interest statement in the “Confidential to Editor” section, and submit your "Accept" recommendation.

Reviewer #2: All comments have been addressed

Reviewer #3: (No Response)

2. Is the manuscript technically sound, and do the data support the conclusions?

Reviewer #2: Yes

Reviewer #3: Yes

3. Has the statistical analysis been performed appropriately and rigorously? 

Reviewer #2: Yes

Reviewer #3: No

4. Have the authors made all data underlying the findings in their manuscript fully available?

Reviewer #2: Yes

Reviewer #3: Yes

5. Is the manuscript presented in an intelligible fashion and written in standard English?

Reviewer #2: Yes

Reviewer #3: Yes

6. Review Comments to the Author

Reviewer #2: (No Response)

Reviewer #3: I repeat, decreased breath sounds (AOR 3.68; 95% CI 0.99-13.76) were not independently associated with radiological pneumonia, please remove it from text! In addition, the result is inaccurate, see the amplitude of the 95%CI.

7. PLOS authors have the option to publish the peer review history of their article (what does this mean?). If published, this will include your full peer review and any attached files.

Reviewer #2: No

Reviewer #3: Yes: Altacilio Nunes

---

## [Author Response · Author response to Decision Letter 4]

2 Jun 2020

RESPONSE TO REVIEWERS` COMMENTS

Reviewers' comments: 

Reviewer's Responses to Questions

Comments to the Author

1. If the authors have adequately addressed your comments raised in a previous round of review and you feel that this manuscript is now acceptable for publication, you may indicate that here to bypass the “Comments to the Author” section, enter your conflict of interest statement in the “Confidential to Editor” section, and submit your "Accept" recommendation.

Reviewer #2: All comments have been addressed

Reviewer #3: (No Response)________________________________________

2. Is the manuscript technically sound, and do the data support the conclusions?

Reviewer #2: Yes

Reviewer #3: Yes________________________________________3. Has the statistical analysis been performed appropriately and rigorously? 

Reviewer #2: Yes

Reviewer #3: No

Action taken: We have checked the statistical analysis. It is correct. ________________________________________

4. Have the authors made all data underlying the findings in their manuscript fully available?

Reviewer #2: Yes

Reviewer #3: Yes________________________________________

5. Is the manuscript presented in an intelligible fashion and written in standard English?

Reviewer #2: Yes

Reviewer #3: Yes________________________________________

6. Review Comments to the Author

Reviewer #2: (No Response)

Reviewer #3: I repeat, decreased breath sounds (AOR 3.68; 95% CI 0.99-13.76) were not independently associated with radiological pneumonia, please remove it from text! In addition, the result is inaccurate, see the amplitude of the 95%CI.

Action taken: We have removed it from the test as suggested by you and have amended the manuscript.

---

## [Decision Letter · Decision Letter 5]

19 Jun 2020

Clinical predictors of radiological pneumonia: a cross-sectional study from a tertiary hospital in Nepal

PONE-D-19-27004R5

Dear Dr. Chaudhary,

We’re pleased to inform you that your manuscript has been judged scientifically suitable for publication and will be formally accepted for publication once it meets all outstanding technical requirements.

Kind regards,

Frederick Quinn

Academic Editor

PLOS ONE

Additional Editor Comments (optional):

Reviewers' comments:

Reviewer's Responses to Questions

**Comments to the Author**

1. If the authors have adequately addressed your comments raised in a previous round of review and you feel that this manuscript is now acceptable for publication, you may indicate that here to bypass the “Comments to the Author” section, enter your conflict of interest statement in the “Confidential to Editor” section, and submit your "Accept" recommendation.

Reviewer #3: All comments have been addressed

2. Is the manuscript technically sound, and do the data support the conclusions?

Reviewer #3: Yes

3. Has the statistical analysis been performed appropriately and rigorously? 

Reviewer #3: (No Response)

4. Have the authors made all data underlying the findings in their manuscript fully available?

Reviewer #3: Yes

5. Is the manuscript presented in an intelligible fashion and written in standard English?

Reviewer #3: Yes

6. Review Comments to the Author

Reviewer #3: Okay, all the reviewers' suggestions/observations have been incorporated into the text. The quality of the manuscript has improved.

7. PLOS authors have the option to publish the peer review history of their article (what does this mean?). If published, this will include your full peer review and any attached files.

Reviewer #3: Yes: Altacilio Nunes

---

## [Editor Report · Acceptance letter]

24 Jun 2020

PONE-D-19-27004R5 

Clinical predictors of radiological pneumonia: a cross-sectional study from a tertiary hospital in Nepal 

Dear Dr. Chaudhary:

I'm pleased to inform you that your manuscript has been deemed suitable for publication in PLOS ONE. Congratulations! Your manuscript is now with our production department. 

Kind regards, 

on behalf of

Dr. Frederick Quinn 

Academic Editor

PLOS ONE